# Effect of Increased IL-1β on Expression of HK in Alzheimer’s Disease

**DOI:** 10.3390/ijms22031306

**Published:** 2021-01-28

**Authors:** Shuangxue Han, Zhijun He, Cornelius Jacob, Xia Hu, Xiao Liang, Wenchang Xiao, Lu Wan, Peng Xiao, Nicola D’Ascenzo, Jiazuan Ni, Qiong Liu, Qingguo Xie

**Affiliations:** 1College of Life Science and Technology, Huazhong University of Science and Technology, Wuhan 430074, China; shuangxuehan@hust.edu.cn (S.H.); cornelius.jacob@student.hs-rm.de (C.J.); hxia0817@hust.edu.cn (X.H.); liangxiao222@gmail.com (X.L.); xiaowc@hust.edu.cn (W.X.); wanlu@hust.edu.cn (L.W.); xiaopeng@hust.edu.cn (P.X.); 2College of Life Sciences and Oceanography, Shenzhen University, Shenzhen 518055, China; hezj@email.szu.edu.cn (Z.H.); jzni@szu.edu.cn (J.N.); liuqiong@szu.edu.cn (Q.L.); 3Department of Medical Physics and Biomedical Engineering, Istituto Neurologico Mediterraneo, Neuromed IRCCS, 86077 Pozzilli, Italy

**Keywords:** HK, IL-1β, Alzheimer’s disease, glucose metabolism, neuroinflammation

## Abstract

Alzheimer’s disease (AD) is a neurodegenerative disease characterized by decreased glucose metabolism and increased neuroinflammation. Hexokinase (HK) is the key enzyme of glucose metabolism and is associated with mitochondria to exert its function. Recent studies have demonstrated that the dissociation of HK from mitochondria is enough to activate the NOD-like receptor protein 3 (NLRP3) inflammasome and leads to the release of interleukin-1β (IL-1β). However, the effect of increased IL-1β on the expression of HK is still unclear in AD. In this paper, we used positron emission tomography (PET), Western blotting and immunofluorescence to study the glucose metabolism, and the expression and distribution of HK in AD. Furthermore, we used lipopolysaccharide (LPS), nigericin (Nig), CY-09 and lonidamine (LND) to treat N2a and N2a-sw cells to investigate the link between IL-1β and HK in AD. The results show decreased expression of HK and the dissociation of HK from mitochondria in AD. Furthermore, a reduction of the expression of IL-1β could increase the expression of HK in AD. These results suggest that inhibiting inflammation may help to restore glucose metabolism in AD.

## 1. Introduction

Alzheimer’s disease (AD) is a common neurodegenerative disease characterized by decreased glucose metabolism and increased neuroinflammation [1,2].

It has been found in several studies that decreased glucose metabolism is associated with the low expression and activity of hexokinase (HK) [3,4], which is the key enzyme in glycolysis. There are four HK isozymes: HK1, HK2, HK3 and HK4. Among them, HK1 and HK2 are known to be associated with the mitochondrial outer membrane and expressed in the brain [5,6,7]. The mitochondrial-HK (mHK) participates in glucose metabolism and regulates the generation of reactive oxygen species (ROS). The dissociation of HK from mitochondria leads to the disruption of the NAD+/NADH ratio, depolarization of membrane potential, and opening of the mitochondrial permeability transition pore. It furthermore promotes the release of mitochondrial ROS and the activation of the NOD-like receptor protein 3 (NLRP3) inflammasome. It also causes the release of pro-apoptotic factors and subsequent apoptosis [8,9,10,11,12]. NLRP3 contains a pyrin domain, a nucleotide-binding site domain and a leucine-rich repeat motif. It interacts with the apoptosis-associated speck-like protein ASC and recruits pro-caspase-1 to form the NLRP3 inflammasome complex. Activation of the NLRP3 inflammasome leads to the activation of caspase-1, and the maturation and release of interleukin-1β (IL-1β) [13,14]. In AD, the NLRP3 inflammasome is activated, and increased IL-1β is implicated in the response to Aβ deposition and contributes to pathology. NLRP3 inflammasome inhibition and IL-1β reduction could enhance the clearance of Aβ and could improve cognitive ability [15].

Although dissociation of HK from mitochondria, NLRP3 inflammation activation and increased IL-1β expression have been found in AD, no relevant studies discuss the link between HK and IL-1β expression in AD [16,17,18]. N2a-sw cells are a murine neuroblastoma cell line stably transfected with the human AβPP-695 Swedish mutation (K595N/M596L). Compared with N2a cells, it can overexpress APP and Aβ. As a model cell line for AD, various characteristics of it have been verified including its increased Aβ levels, increased expression of cytochrome c, increased IL-1β level, increased ratio of Bax to Bcl-2, reduced autophagy and so on [19,20,21]. N2a-sw cells are important model cells in the in vitro study of AD.

In this paper, we aimed at determining the expression and distribution of HK and at studying the effect of increased IL-1β on the expression and distribution of HK in AD. We first detected the overall expression and distribution of HK1 and HK2. Then, we detected the expression of NLRP3, pro-caspase-1, caspase-1, pro-IL-1β and IL-1β. At last, we used lipopolysaccharide (LPS), nigericin (Nig), CY-09 and lonidamine (LND) to treat N2a and N2a-sw cells to investigate the link between IL-1β and HK in AD.

## 2. Results

### 2.1. Decreased Glucose Metabolism in Triple-Transgenic (3 × Tg) AD Mice

To compare the glucose metabolism levels in non-transgenic (NTg) mice and triple-transgenic (3 × Tg) AD mice, we detected ^18^F-flurodeoxyglucose (^18^F-FDG) uptake using positron emission tomography (PET). ^18^F-FDG is an analog of glucose and can also be catalytically processed by HK. However, it cannot be catalytically processed by glucose-phosphate isomerase and cannot leave cells, so its uptake represents the glucose metabolism level [3]. The distribution of the ^18^F-FDG uptake in the two types of mice is shown in Figure 1a. As reported in Figure 1b, we observed that the average standardized uptake values (SUVs) measured in the whole brain, cortex and hippocampus were lower in 3 × Tg AD mice than in NTg mice, with a *p*-value lower than 0.05. These results indicate decreased glucose metabolism in 3 × Tg AD mice.

### 2.2. Decreased Expression of HK In Vivo and In Vitro

HK is the key enzyme in glucose metabolism. The decreased glucose metabolism in 3 × Tg AD mice may imply the abnormal expression and distribution of HK. HK1 and HK2 are known to be mitochondrial hexokinase isotypes because they participate in glucose metabolism by binding to mitochondria. They are mainly expressed in mitochondria, but there is also a small expression in the cytoplasm [5,6,7]. To study the expression of HK, we first studied the mRNA and overall protein levels of HK1 and HK2 using quantitative real-time PCR (Q-PCR) and Western blotting, respectively, in vivo and in vitro.

As shown in Figure 2a, the mRNA levels of *HK1* in 3 × Tg AD mice were lower than in NTg mice. We also found a reduction in N2a-sw cells of *HK1* with a *p*-value lower than 0.001 (Figure 2d). The protein expression of HK1 was consistent with the results for the mRNA level, with a *p*-value lower than 0.05 in the 3 × Tg AD mice (Figure 2b,c) and N2a-sw cells (Figure 2e,f). We also detected the expression of HK2 and found a decrease in vivo and in vitro, with no significant difference in mRNA and protein levels. A significant decrease in hexokinase activity was found in N2a-sw cells (Figure 2g). These results show that decreased expression of HK existed in AD.

### 2.3. Dissociation of HK from Mitochondria In Vivo and In Vitro

HK1 and HK2 are known to be associated with mitochondria and catalyze glucose. They cannot exert their function once they dissociate from mitochondria. We had detected the low expression of HK in AD, and then, we studied the distribution of HK1 and HK2 in vivo and in vitro. Because we did not obtain a suitable antibody for HK1 and HK2 for immunohistochemistry, we isolated the mitochondria from NTg mice and 3 × Tg AD mice to detect the protein expression of HK1 and HK2 in the mitochondria. The results are shown in Figure 3a,b; decreased protein expression of HK1 and HK2 was detected in the mitochondria, and increased protein expression of HK1 and HK2 was found in the cytoplasm. In comparison with NTg mice, mitochondrial HK1 (mHK1) expression and cytoplasmic HK1 (cHK1) expression were statistically significantly different, with *p*-values lower than 0.01, in 3 × Tg AD mice.

Furthermore, we studied the distribution of HK1 and HK2 in vitro by immunofluorescence. As the immunofluorescence assay was performed after the transfection of Mito-DsRed, the immunofluorescence signals of the mitochondrial network with Mito-DsRed were a little weak. Moreover, the bleed-through effects lead to influence from HK and DAPI on Mito-DsRed. However, these issues do not make a significant contribution to the results [22]. As reported in Figure 3c,d, compared with in N2a cells, the Pearson’s coefficient of the correlation between HK1 and mitochondria was lower in N2a-sw cells, with a p-value lower than 0.001. The Pearson’s coefficient of the correlation between HK2 and mitochondria was also lower in N2a-sw cells but with no significance. There was no difference in the coefficient of the overlap of HK1 and HK2 with mitochondria in N2a-sw cells compared with in N2a cells. These results demonstrate an abnormal distribution of HK on mitochondria in AD, further indicating that decreased expression of mHK correlates not only with lower overall expression of HK, but also with its dissociation from mitochondria in AD.

### 2.4. Increased Protein Expression of IL-1β In Vivo and In Vitro

Previous researchers found the dissociation of HK from mitochondria to activate the NLRP3 inflammasome and lead to an increased release of IL-1β in bone marrow-derived macrophages [6]. However, there are no relevant reports in N2a and N2a-sw cells. We had confirmed the dissociation of HK1 and HK2 in vivo and in vitro. We detected the protein expression of NLRP3, pro-caspase-1, cleaved-caspase-1 (P20), pro-IL-1β and IL-1β in vivo and in vitro using Western blotting.

As shown in Figure 4a–c, the mRNA level and protein level of NLRP3 were significantly increased in 3 × Tg AD mice, with *p*-values lower than 0.05. The protein expression of P20 and IL-1β was also higher in 3 × Tg AD mice, with *p*-values lower than 0.05. However, the expression of pro-IL-1β decreased in 3 × Tg AD mice, with a *p*-value lower than 0.05. There was no significant difference in the expression of pro-caspase-1 between NTg and 3 × Tg AD mice. The expression in the N2a-sw cells (Figure 4d–f) was consistent with the results in vivo. However, there were no statistically significant differences in the mRNA level of NLRP3 and protein level of pro-IL-1β. These results imply the activation of the NLRP3 inflammasome and increased protein expression of IL-1β in AD.

### 2.5. Decreased HK Causes the Increase in IL-1β in LND-Treated N2a Cells

Preliminary results show that the decreased expression and abnormal distribution of HK were accompanied by increased expression of IL-1β in AD. Decreased mHK has shown a correlation with the dissociation from mitochondria and the diminution of the overall expression of HK. Here, we mainly focus on the relationship between the decreased overall expression and association of HK with increased expression of IL-1β in AD, as increased IL-1β expression and decreased HK expression had been found in N2a-sw cells, model cells for AD. Therefore, we used LND and LPS + Nig + CY-09 to only treat N2a cells to study the relationship between HK and IL-1β. We then used CY-09 only to treat N2a-sw cells to confirm the relationship with AD.

We first treated the N2a cells with lonidamine (LND), an inhibitor of the expression and association with mitochondria of HK. As shown in Figure 5a–e, the expression of HK1 was decreased with increased treatment time and dose in LND-treated N2a cells. The differences were significant compared with untreated N2a cells, with *p*-values lower than 0.05, 0.01 and 0.01, respectively, in the 25 µM-12 h-treated, 50 µM-6 h-treated and 50 µM-12 h-treated groups. The expression of HK2 was also decreased in LND-treated N2a cells but showed no significant difference compared with untreated N2a cells. The expression of pro-IL-1β was decreased in 25 µM-6 h-treated, 50 µM-6 h-treated and 50 µM-12 h-treated cells and with a *p*-value lower than 0.05 in the 50 µM-12 h-treated group. However, the expression of IL-1β was increased with increased treatment time and dose in LND-treated N2a cells. The differences were also significant compared with untreated N2a cells, with *p*-values lower than 0.05, in the 25 µM-12 h-treated, 50 µM-6 h-treated and 50 µM-12 h-treated groups. Increased IL-1β secretion was also found in LND-treated N2a cells by ELISA (Figure 5f). These results indicate that the expression of IL-1β was affected by HK in N2a cells.

### 2.6. Increased IL-1β Leads to the Reduction of HK in LPS-, Nig- and CY-09-Treated N2a Cells

The expression of IL-1β was affected by HK. Then, we investigated whether the expression of HK was also affected by IL-1β. We treated the N2a cells with lipopolysaccharide (LPS), nigericin (Nig) and CY-09. LPS and Nig are activators of the NLRP3 inflammasome. The activation of the NLRP3 inflammasome leads to the increased expression of IL-1β [23]. The dissociation of HK from mitochondria can activate the NLRP3 inflammasome, but there are no relevant studies to prove correlations between HK and AIM2, NLRC4 and Pyrin. To avoid the influence of other inflammasomes, we chose CY-09 to treat the LPS- and Nig-treated cells, which is a specific inhibitor of the NLRP3 inflammasome. It reduces the expression of IL-1β by inhibiting NLRP3 inflammasome activation.

As reported in Figure 6a–e, compared with untreated N2a cells, the protein expression of pro-IL-1β and IL-1β was remarkably increased in LPS-treated N2a cells, with a *p*-value lower than 0.05 (Lane 3). Similarly, the expression of IL-1β was also significantly increased in LPS + Nig-treated N2a cells (Lane 5), with a *p*-value lower than 0.001, compared with untreated N2a cells. However, it decreased after the CY-09 treatment (Lane 6), with a *p*-value lower than 0.01, compared with LPS + Nig-treated N2a cells. The expression of HK1 and HK2 was lower in LPS-treated N2a cells than untreated N2a cells and the lowest in the LPS + Nig-treated N2a cells, with *p*-values lower than 0.01. Their expression increased after CY-09 treatment but not significantly so compared with in LPS + Nig-treated N2a cells (Lanes 4 and 6). Increased secretion of IL-1β was found in LPS- and LPS + Nig-treated N2a cells, with *p*-values lower than 0.05. Significantly decreased secretion of IL-1β was detected after CY-09 treatment (Figure 6f). Although the protein expression of IL-1β was increased in CY-09-treated N2a cells (Figure 6a,e, Lane 2), there were no statistical differences compared with N2a cells. The results of the ELISA confirmed the inhibition by CY-09 of the expression of IL-1β. All the results suggest that the expression of HK was also affected by IL-1β in N2a cells.

### 2.7. Increased Expression of HK in CY-09-Treated N2a-sw Cells

Reduced expression of IL-1β leads to an increase in HK1 in LPS + CY-09- and LPS + Nig + CY-09-treated N2a cells, and increased expression of IL-1β had been found in N2a-sw cells. Hence, we used CY-09 to treat N2a-sw cells and detected the expression of HK in CY-09-treated N2a-sw cells. As shown in Figure 7a–e, compared with untreated N2a-sw cells, the expression of pro-IL-1β and IL-1β was lower in CY-09-treated N2a-sw cells (Lanes 2–4), with *p*-values lower than 0.05 and 0.01, respectively, in the 10 µM-2 h-treated and 10 µM-4 h-treated groups. The secretion of IL-1β was also reduced in CY-09-treated N2a-sw cells and with a *p*-value lower than 0.05 in 10 µM-4 h-treated N2a-sw cells (Figure 7f). However, the expression of HK1 was increased in CY-09-treated N2a-sw cells, and there was a significant difference between untreated N2a-sw cells and N2a-sw cells treated with 10 μM CY-09 for 4 h, with a *p*-value lower than 0.01. The expression of HK2 was also increased in N2a-sw cells treated with 5 μM and 10 μM CY-09 for 4 h, but no significance was found. The results confirm that the expression of HK was increased by reducing the expression of IL-1β in AD.

### 2.8. Decreased IL-1β Can Improve the Abnormal Distribution in Treated Cells

The detachment of HK from mitochondria induces the activation of the NLRP3 inflammation and the release of IL-1β. We found that a reduction of the expression of IL-1β can increase the expression of HK1. Then, we used immunofluorescence to study if the reduction of IL-1β expression could also improve the abnormal distribution of HK1 in cells.

As shown in Figure 8, the Pearson’s correlation coefficient decreased in LND-treated N2a cells and increased in CY-09-treated N2a-sw cells, with *p*-values lower than 0.05. We also observed the distribution of HK1 in LPS-, Nig- and CY-09-treated N2a cells. As reported in Figure 9, we found that in LPS- and LPS + Nig-treated N2a cells, the Pearson’s correlation coefficient was lower than for N2a cells. We also observed a significant decrease in LPS + Nig-treated cells, with a *p*-value lower than 0.05. In comparison with LPS- and LPS + Nig-treated cells, the Pearson’s correlation coefficient was higher in LPS + CY-09- and LPS + Nig + CY-09-treated N2a cells. There was no difference in the overlap coefficient between each group. These results show that the abnormal distribution of HK can improved by suppressing the inflammation and decreasing IL-1β levels.

### 2.9. Decreased ROS Levels and Increased HK Activity in CY-09-Treated N2a-sw Cells

Finally, we detected the hexokinase activity, mitochondrial membrane potential, ROS level and Aβ oligomers expression in cells treated with 10 µM CY-09 for 4 h. As shown in Figure 10, the hexokinase activity in CY-09-treated N2a-sw cells increased significantly, with a *p*-value lower than 0.01. Additionally, the ROS level decreased in CY-09-treated N2a-sw cells, with a *p*-value lower than 0.05. However, there was no statistical difference in mitochondrial membrane potential. We also examined the levels of Aβ oligomers by Western blotting using the specific antibody 6E10 in N2a-sw cells and cells treated with 10 µM CY-09 for 4 h. As shown in Figure 10d–e, CY-09 notably decreased the production of Aβ oligomers (pentamers, hexamers, heptamers, 14-mers, 17-mers and 26-mers), with *p*-values lower than 0.05, 0.05, 0.01, 0.01, 0.01 and 0.001, respectively. The trimers and decamers were remarkably increased, with *p*-values lower than 0.001. Overall, these results prove that a suppression of the inflammatory response is helpful for restoring hexokinase activity, decreasing Aβ oligomers and protecting cells from apoptosis.

## 3. Discussion

In this study, we detected the decreased expression and activity of HK and confirmed the release of HK from mitochondria in 3 × Tg AD mice and N2a-sw cells. Moreover, we found that it is possible to reduce the expression of IL-1β, increase the co-localization of HK and mitochondria and reduce ROS levels by suppressing the activation of the NLRP3 inflammasome in AD.

HK is combined with mitochondria to participate in glucose metabolism and the regulation of ROS generation [10]. In AD, Aβ binds to mitochondria and disrupts the combination of HK with mitochondria, inducing the dissociation of HK [9]. A decreased expression of HK and dissociation from mitochondria reduces glucose metabolism, promotes mitochondrial dysfunction, and leads to a disruption of the NAD+/NADH ratio, the depolarization of membrane potential and the opening of mitochondrial permeability transition pores. It further promotes mitochondrial reactive oxygen species (ROS) release and NLRP3 inflammasome activation [7,8,9,10,11,12]. Consistently with these observations, we found decreased expression of HK and combination of it with mitochondria in N2a-sw cells. Moreover, we detected increased NLRP3, cleaved caspase-1, and IL-1β in N2a-sw cells. NLRP3-mediated caspase-1 activation can promote GAPDH proteolysis, resulting in decreased glycolytic potential in aging and further limiting glucose metabolism in the aging process [15]. That indicates that the activation of NLRP3 may also influence the expression and activity of HK. We used CY-09 to inhibit the activation of NLRP3 and expression of IL-1β in N2a-sw cells. Our results indicate that, in CY-09-treated N2a-sw cells, the expression and secretion of IL-1β decreased. A recovery of the expression, association and activity of HK was found in CY-09-treated N2a-sw cells. We also detected a reduction of the ROS level in CY-09-treated N2a-sw cells, which confirmed the regulation of ROS generation by mHK. Our results demonstrate that increased IL-1β can reduce the expression of HK and combination of it with mitochondria in AD.

Suppressing the activation of the NLRP3 inflammasome and expression of IL-1β can increase the expression of HK and combination of it with mitochondria in AD. Increased HK1 can raise the glycolytic rate and glucose uptake and promote the recovery of cognition performance in AD [24]. High expression of HK1 can prevent the neuroinflammation elicited by advanced glycation end products (AGEs), which originate from the spontaneously reaction of fructose with proteins in the brain [25]. Moreover, the inhibition of NLRP3 inflammasome activation and IL-1β expression can also enhance the clearance of Aβ and improve cognitive abilities in AD [15]. We detected the expression of Aβ oligomers in N2a-sw cells and CY-09-treated N2a-sw cells. Though there was an increase in trimers and decamers, decreased expression of pentamers, hexamers, heptamers, 14-mers, 17-mers and 26-mers was found in CY-09-treated N2a-sw cells. Moreover, considering that neuroinflammation has been recognized as a new target for the treatment of AD [14,15], our data provide new evidence for this therapy’s potential.

In this paper, we also used LND, LPS, Nig and CY-09 to treat N2a cells to investigate the relationship between the expression of HK and IL-1β. We first used LND to disrupt the expression and distribution of HK. In agreement with previous studies [6], decreased expression and abnormal distribution of HK led to the increased expression and secretion of IL-1β. Then we used LPS, Nig and CY-09 to examine the influence of increased IL-1β on HK. The results showed that decreased expression and combination of HK was detected in cells with increased IL-1β, and they were improved after adding CY-09. Those results show that the expression of HK and the expression of IL-1β affect each other. However, a high positive correlation between glucose metabolism and neuroinflammation in 6-month-old AD model mice but not in 15-month-old mice was pointed out in previous studies [26]. It was considered to be, perhaps, caused by the compensatory increase in metabolism at disease onset.

In conclusion, our study confirmed a reduction of HK expression and activity and abnormal distribution in AD. We preliminarily explored the relationship between decreased HK and increased IL-1β in AD. We will further investigate the mechanism of the interaction between HK and IL-1β.

## 4. Materials and Methods

### 4.1. Animals

We used 9-month-old female non-transgenic (NTg) mice and triple-transgenic AD (3 × Tg AD) model mice in the experiments. The NTg mice were purchased from Beijing Huafukang Biotechnology Co., Ltd. The 3 × Tg AD mice were purchased from the Jackson Laboratory, and they harbor the mutated human genes APP (SWE), PS1 (M146V) and Tau (P301L) under the control of Thy1.2 promoters. All the mice were housed at 22 °C and with a 12 h light/night cycle with free access to food and water [27].

All the animal experiments complied with the Animal Care and Institutional Ethical Guidelines in China and were approved by the Institutional Animal Care and Use Committee of Tongji Medical College of Huazhong University of Science and Technology (IACUC Number: 2390; Approved Date: 27 February 2018).

### 4.2. Positron Emission Tomography (PET) Scanning

We scanned all the mice with ^18^F-FDG-PET in static scan mode. After 12–16 h of fasting, we injected the mice with 200 μCi ^18^F-FDG in veins. We acquired a 10 min-long PET scan 60 min after injection (Trans-PET^®^ BioCaliburn^®^ 700 Raycan Technology Co., Ltd., Suzhou, China). We reconstructed the PET data using the 3D OSEM algorithm. We quantified the standard uptake values (SUVs) of the whole brain, cortex and hippocampus.

### 4.3. Cell Culture and Treatment

We cultured N2a and N2a-sw cells. N2a-sw cells are a murine neuroblastoma cell line stably transfected with the human AβPP-695 Swedish mutation (K595N/M596L). We cultured these cells in 45% DMEM (Hyclone, USA) and 55% Opti-MEM (Gibco, USA) containing 5% fetal bovine serum (FBS). For the N2a-sw cells, we supplemented 0.2 mg/mL G-418 into the culture medium to maintain genetic stability. We cultured all these cells in a humidified atmosphere at 37 °C with 5% CO_2_ [20].

To observe the impact of HK on IL-1β, we used lonidamine (LND) to inhibit HK expression. We selected LND at different concentrations (25 and 50 μM) to treat N2a cells for 6 and 12 h, respectively. To investigate the influence of IL-1β on HK, we treated the N2a cells with 1 μM LPS for 3 h, 10 μM CY-09 for 1 h and 10 μM nigericin (Nig) for 30 min, to induce or inhibit NLRP3 inflammasome activation, following Hua Jiang’s description [28]. We divided the cells into six groups: the untreated group, LPS-treated group, CY-09-treated group, LPS + CY-09-treated group, LPS + Nig-treated group and LPS + Nig + CY-09-treated group. At last, we used varying doses (5 and 10 μM) and treatment times (2 and 4 h) for CY-09 to treat the N2a-sw cells to study the effect of IL-1β on HK in AD.

### 4.4. Quantitative Real-Time PCR (Q-PCR) Assay

We extracted the total RNA from cells and tissues using the RNA fast200 kit (Fastagen, Shanghai, China) and quantified it using a NanoDrop 2000 (Thermo Fisher Scientific, Waltham, MA, USA). Then, we reverse transcribed the RNA into complementary DNA using the HiScript^®^ III RT SuperMix for q-PCR (+gDNA wiper) kit (Vazyme Biotech Co., Ltd., Nanjing, China) following the manufacturer’s instructions. We performed Q-PCR using the 2 × ChamQ SYBR Color qPCR Master Mix (High ROX Premixed) kit (Vazyme Biotech Co., Ltd., Nanjing, China) on the Applied Biosystems QuantStudio™ 6 & 7 Real-Time PCR System (Thermo Fisher Scientific, Waltham, MA, USA). The thermal cycling protocol included initial heating at 95 °C for 2 min followed by 40 cycles of 95 °C for 10 s, 60 °C for 34 s, and 72 °C for 30 s. The following primer sequences were used: *HK1* (GenBank Accession Number: NM_001146100), *HK2* (GenBank Accession Number: NM_013820), *NLRP3* (GenBank Accession Number: NM_001127462) and *β-actin* (GenBank Accession Number: NM_007393).

### 4.5. Brain Mitochondrion Isolation

Mitochondria were isolated using a mitochondrial isolation kit (Thermo Fisher Scientific, Waltham, MA, USA) following the manufacturer’s instructions. Briefly, brain tissues of the NTg and 3 × Tg AD mice were collected and homogenized using a glass grinding rod, and then centrifuged at 700× *g* for 10 min at 4 °C. The supernatant was further centrifuged at 12,000× *g* for 15 min at 4 °C. The supernatant (cytoplasm) was collected and stored at −80 °C. The pellet (mitochondria) was also stored at −80 °C after washing.

### 4.6. Western Blotting

We sonicated cells and mitochondria in the buffer for the radio-immunoprecipitation assay (RIPA) containing a protease inhibitor cocktail (Sigma, St. Louis, MO, USA). We collected the supernatant after centrifuging it at 12,000× *g* for 20 min. We determined the protein concentrations using a BCA protein assay kit (Thermo Fisher Scientific, Waltham, MA, USA). We electrophoresed equal amounts of total protein lysates (20 µg per well) for each sample on SDS-PAGE gels, and we transferred the proteins to polyvinylidene fluoride (PVDF) membranes (Millipore, Billerica, MA, USA). Then, we blocked the membranes with 5% fat-free milk diluted in Tris-buffered saline Tween-20 (TBST) for 2 h, followed by overnight incubation with primary antibodies: anti-HK1 (1:1000, Abcam, Cambridge, UK), anti-HK2 (1:1000, Proteintech), anti-NLRP3 (1:1000, AdipoGen, Epalinges, Switzerland), anti-caspase-1 (P20) (1:1000, AdipoGen, Epalinges, Switzerland), anti-pro-IL-1β (1:1000, Abcam, Cambridge, UK), anti-IL-1β (1:2500, Abcam, Cambridge, UK), anti-β-actin (1:2500, Abways, Riga, Latvia) and anti-VDAC1 (1:1000, Abcam, Cambridge, UK). The next day, we incubated the membranes with secondary antibodies conjugated with horseradish peroxidase (HRP) for 2 h at room temperature. We visualized the immunoreactivity using the KODAK Image Station 4000 MM (Carestream Health Inc., New Haven, CT, USA). We analyzed all band intensities using the ImageJ software (NIH, Bethesda, MD, USA).

### 4.7. ELISA Assay

We used the Mouse IL-1β ELISA kit (purchased from NEOBIOSCIENCE) to detect the secretion of IL-1β. All the experiments were conducted following the product manual.

### 4.8. Hexokinase Activity Assay

We used the Micro Hexokinase (HK) Assay Kit (purchased from Beijing Solarbio Science & Technology Co., Ltd., Beijing, China) to detect the hexokinase activity. We carried out the experiments by following the manufacturer’s instructions.

### 4.9. Mitochondrial Membrane Potential and ROS Assay

We used kits (purchased from Beyotime Biotechnology Co., Ltd., Beijing, China) to study the mitochondrial membrane potential and ROS levels. When seeded cells were 85% confluent, we used 10 μM CY-09 to treat N2a-sw cells for 4 h. Then, we performed the experiments following the manufacturer’s instructions. The relative fluorescence intensity (Red/Green) was calculated to determine the mitochondrial membrane potential. The fluorescence intensity quantitatively represented the ROS levels.

### 4.10. Transfection Assay

We visualized the cell mitochondria by the transfection of pDsRed2-Mito (TAKARA, Kusatsu, Japan). We performed the transfection assay using Lipofectamine 2000 according to the manufacturer’s protocol when seeded cells were 70% confluent. One day before transfection, we replaced the culture medium with a medium without antibiotics. For each transfection sample, we diluted DNA and Lipofectamine 2000 in 250 μL of culture medium without antibiotics, separately. After 5 min of incubation, the diluted DNA was combined with diluted Lipofectamine 2000 (total, 500 μL), mixed gently and incubated for 20 min at room temperature. Then, we added 500 μL of complexes to each sample and mixed gently. After 4–6 h, we replaced the medium with a culture medium without antibiotics. We cultured the cells for 48 h, and then, we used them for subsequent experiments.

### 4.11. Immunofluorescence

We performed the assay following a previously established protocol [27]. Briefly, we fixed N2a and N2a-sw cells in 4% paraformaldehyde for 20 min, and then, we washed them three times in phosphate-buffered saline (PBS). After incubating in 0.2% Triton X-100 for 20 min and blocking with 10% goat serum in PBS for 1 h, we incubated the cells with primary antibodies including HK1 (1:200) and HK2 (1:200) overnight at 4 °C. The next day, we washed them and incubated them with secondary antibodies (1:1000) labeled with Dylight-488 for 1 h. We captured images with a Multiphoton Microscopy LSM 710 NLO (Carl Zeiss AG, Berlin, Germany), and we analyzed them using the ImageJ software.

### 4.12. Statistical Analysis

In this study, we used a semi-quantitative analysis method to quantify the mean SUV of the whole brain, cortex and hippocampus from the PET images. We calculated the SUV using the following equation:(1)$$SUV=\frac{C_{T}}{D_{Inj}}\cdot W_{S}$$

*C_T_* is the radioactivity in the region of interest, with the unit of mCi/g tissue. *D_Inj_* is the injected dose of radioactivity, with the unit of mCi. *W_S_* is the weight of the mouse, with the unit of g.

We used the 2^−ΔΔCT^ method to determine the relative mRNA expression of *HK1*, *HK2* and *NLRP3*. We obtained the cycle threshold (CT) from the Q-PCR results. We used the following formulas to determine the values of ΔΔCT: ΔCT = CT_target gene_ − CT_β-actin_. ΔΔCT = ΔCT_N2a-sw cells_ − ΔCT_N2a cells_ for cells and ΔΔCT = ΔCT_3 × Tg_
_AD mice_ − ΔCT_N__Tg mice_ for mice. We used grayscale analysis to quantify the protein expression. We measured the grayscale values of each band in the Western blot images using the ImageJ software. We normalized the grayscale values of the target protein to the grayscale values of β-actin or VDAC1. Immunofluorescence co-localization analysis was used to examine the distribution of HK1 and HK2 in mitochondria.

We present all the data as mean ± SD, and they were analyzed with Graphpad Prism (GraphPad Inc., La Jolla, CA, USA). We performed the statistical analysis of two-group differences using unpaired, parametric two-tailed Student *t*-test with no Welch’s correction.

## Figures and Tables

**Figure 1 ijms-22-01306-f001:**
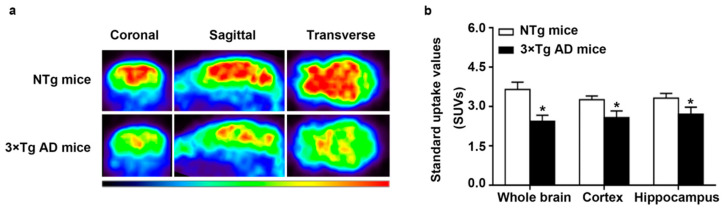
^18^F-FDG uptake in non-transgenic (NTg) and triple-transgenic (3 × Tg) Alzheimer’s disease (AD) mice. (**a**) shows coronal, sagittal and transverse images, and (**b**) presents the standardized uptake values (SUVs) in the whole brain, cortex and hippocampus in NTg and 3 × Tg AD mice. (*n* = 6, mean ± SD, Student’s *t*-test, * *p* < 0.05 vs. NTg mice).

**Figure 2 ijms-22-01306-f002:**
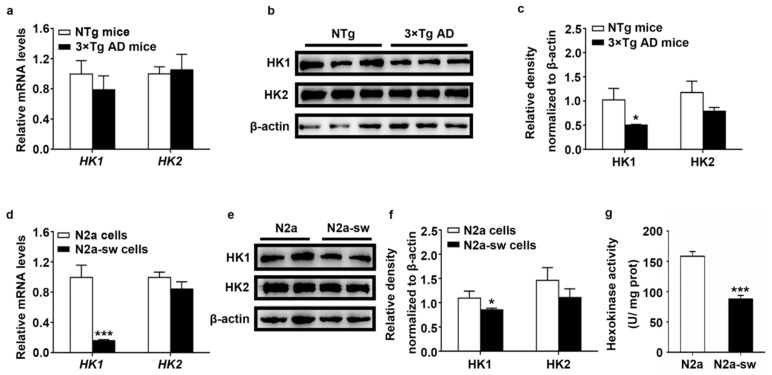
Expression of HK1 and HK2 in vivo and in vitro. (**a**) Transcriptional activities of HK1 and HK2 were measured by Q-PCR in NTg and 3 × Tg AD mice. Quantification was analyzed by the 2^−ΔΔCT^ method. (*n* = 6, mean ± SD, Student’s *t*-test). (**b**,**c**) Western blot detection of HK1 and HK2 in NTg and 3 × Tg AD mice. (*n* = 6, mean ± SD, Student’s *t*-test, * *p* < 0.05 vs. NTg mice). (**d**) Transcriptional activities of HK1 and HK2 were measured by Q-PCR in N2a and N2a-sw cells. Quantification was analyzed by the 2^−ΔΔCT^ method. (*n* = 3, mean ± SD, Student’s *t*-test, *** *p* < 0.001 vs. N2a cells). (**e**,**f**) Western blot detection of HK1 and HK2 in N2a and N2a-sw cells. (*n* = 3, mean ± SD, Student’s *t*-test, * *p* < 0.05 vs. N2a cells). (**g**) Hexokinase activity in N2a and N2a-sw cells. (*n* = 3, mean ± SD, Student’s *t*-test, *** *p* < 0.001 vs. N2a cells).

**Figure 3 ijms-22-01306-f003:**
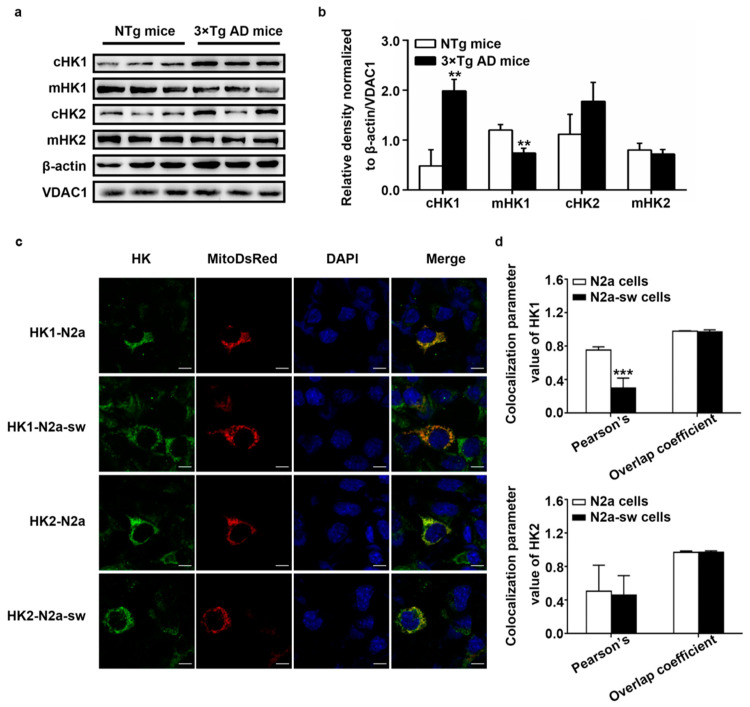
Distribution of HK1 and HK2 in vivo and in vitro. (**a**,**b**) Western blot detection of HK1 and HK2 expression in cytoplasm and mitochondria in NTg and 3 × Tg AD mice. cHK1 and cHK2 were normalized to β-actin. mHK1 and mHK2 were normalized to VDAC1. (*n* = 6, mean ± SD, Student’s *t*-test, ** *p* < 0.01 vs. NTg mice). (**c**) Immunofluorescence staining of HK1 and HK2; Mito-DsRed-transfected N2a and N2a-sw cells were imaged. DAPI is a nuclear stain. (**d**) Co-localization analysis of HK1 and HK2 with mitochondria in N2a and N2a-sw cells. Pearson’s correlation and overlap coefficient were analyzed. (*n* = 6, mean ± SD, Student’s *t*-test, *** *p* < 0.001 vs. N2a cells.)

**Figure 4 ijms-22-01306-f004:**
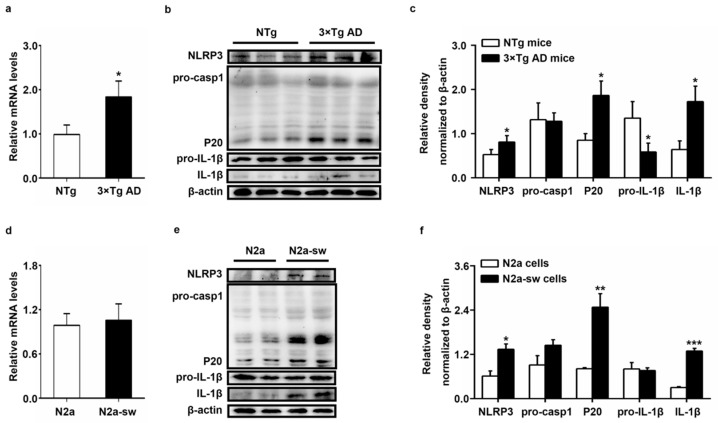
Expression of NLRP3, pro-caspase-1, cleaved-caspase-1 (P20), pro-IL-1β and IL-1β in vivo and in vitro. (**a**) Transcriptional activities of NLRP3 were measured by Q-PCR in NTg and 3 × Tg AD mice. Quantification was analyzed by the 2^−ΔΔCT^ method. (*n* = 6, mean ± SD, Student’s *t*-test, * *p* < 0.05 vs. NTg mice). (**b**,**c**) Western blot detection of NLRP3, pro-caspase-1, P20, pro-IL-1β and IL-1β expression in NTg and 3 × Tg AD mice. (*n* = 6, mean ± SD, Student’s *t*-test, * *p* < 0.05 vs. NTg mice). (**d**) Transcriptional activities of NLRP3 were measured by Q-PCR in N2a and N2a-sw cells. Quantification was analyzed by the 2^−ΔΔCT^ method. (*n* = 3, mean ± SD, Student’s *t*-test). (**e**,**f**) Western blot analysis of NLRP3, pro-caspase-1, P20, pro-IL-1β and IL-1β expression in N2a and N2a-sw cells. (*n* = 3, mean ± SD, Student’s *t*-test; * *p* < 0.05, ** *p* < 0.01, *** *p* < 0.001 vs. N2a cells).

**Figure 5 ijms-22-01306-f005:**
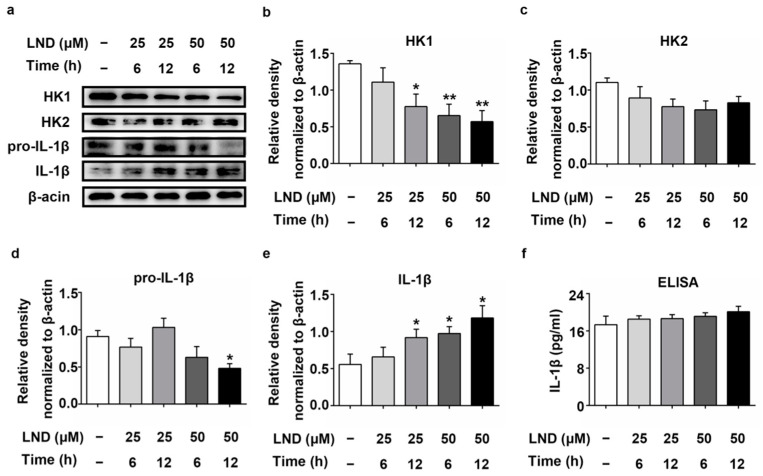
Expression of HK1, HK2, pro-IL-1β and IL-1β in lonidamine (LND)-treated N2a cells. (**a**–**e**) Western blot analysis of HK1, HK2, pro-IL-1β and IL-1β in LND-treated N2a cells. (*n* = 3, mean ± SD, Student’s *t*-test; * *p* < 0.05, ** *p* < 0.01 vs. untreated N2a cells). (**f**) Detection of IL-1β secretion by ELISA in LND-treated N2a cells (*n* = 3, mean ± SD, Student’s *t*-test. “−” means no treatment).

**Figure 6 ijms-22-01306-f006:**
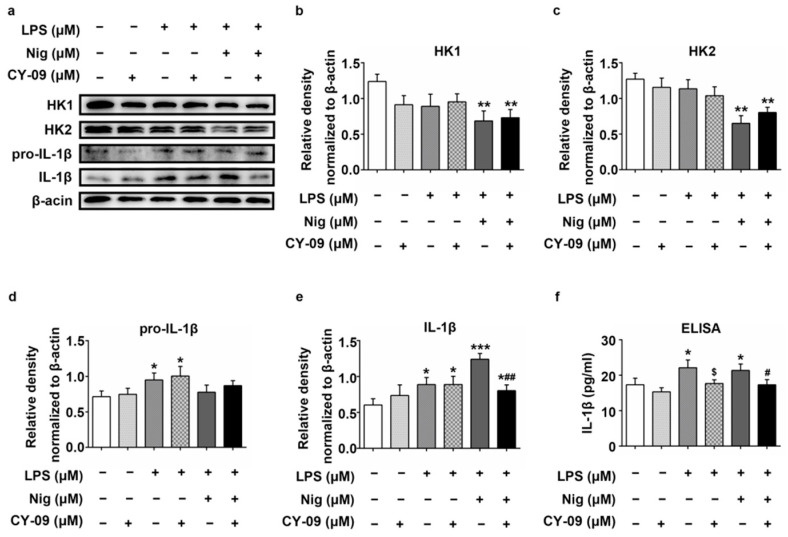
Expression of HK1, HK2, pro-IL-1β and IL-1β in treated N2a cells. (**a**–**e**) Western blot analysis of HK1, HK2 pro-IL-1β and IL-1β in LPS-, Nig- and CY-09-treated N2a cells. (*n* = 3, mean ± SD, Student’s *t*-test; * *p* < 0.05, ** *p* < 0.01, *** *p* < 0.001 vs. untreated N2a cells; ^##^
*p* < 0.01 vs. LPS + Nig-treated N2a cells). (**f**) Detection of IL-1β secretion by ELISA in treated N2a cells. (*n* = 3, mean ± SD, Student’s *t*-test, * *p* < 0.05 vs. untreated N2a cells, ^$^
*p* < 0.05 vs. LPS-treated N2a cells, ^#^
*p* < 0.05 vs. LPS + Nig-treated N2a cells. “−” means not added in the treatment. “+” means added in the treatment).

**Figure 7 ijms-22-01306-f007:**
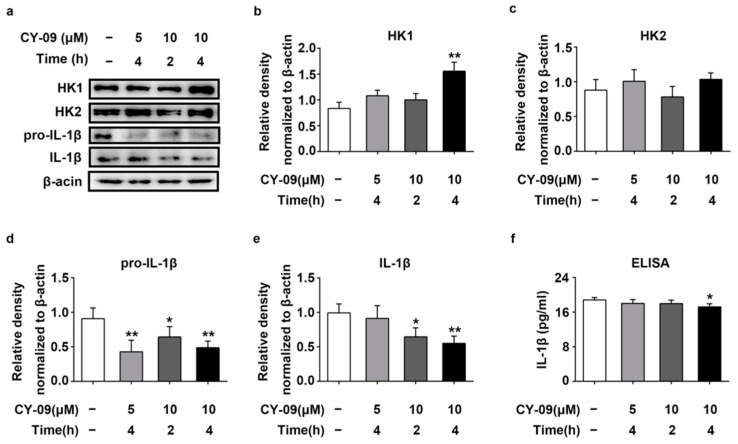
Expression of HK1, HK2, pro-IL-1β and IL-1β in CY-09-treated N2a-sw cells. (**a**–**e**) Western blot analysis of HK1, HK2, pro-IL-1β and IL-1β in CY-09-treated N2a-sw cells. (*n* = 3, mean ± SD, Student’s *t*-test; * *p* < 0.05, ** *p* < 0.01 vs. untreated N2a-sw cells). (**f**) ELISA detection of the IL-1β secretion in CY-09-treated N2a-sw cells (*n* = 3, mean ± SD, Student’s *t*-test, * *p* < 0.05 vs. untreated N2a-sw cells. “−” means no treatment).

**Figure 8 ijms-22-01306-f008:**
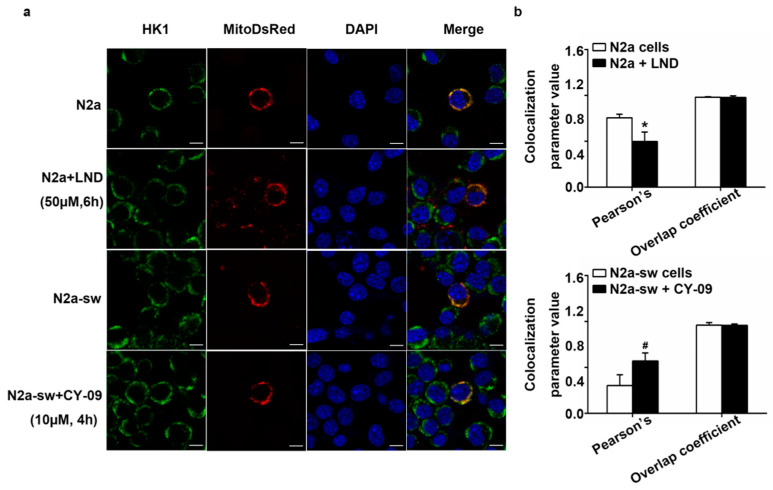
Distribution of HK1 in treated cells. (**a**) Immunofluorescence staining of HK1; Mito-DsRed-transfected N2a cells, LND-treated N2a cells, N2a-sw cells and CY-09-treated N2a-sw cells were imaged. DAPI is a nuclear stain. (**b**) Co-localization analysis of HK1 with mitochondria in cells. Pearson’s correlation and overlap coefficients were analyzed. (*n* = 6, mean ± SD, Student’s *t*-test, * *p* < 0.05 vs. N2a cells, ^#^
*p* < 0.05 vs. N2a-sw cells).

**Figure 9 ijms-22-01306-f009:**
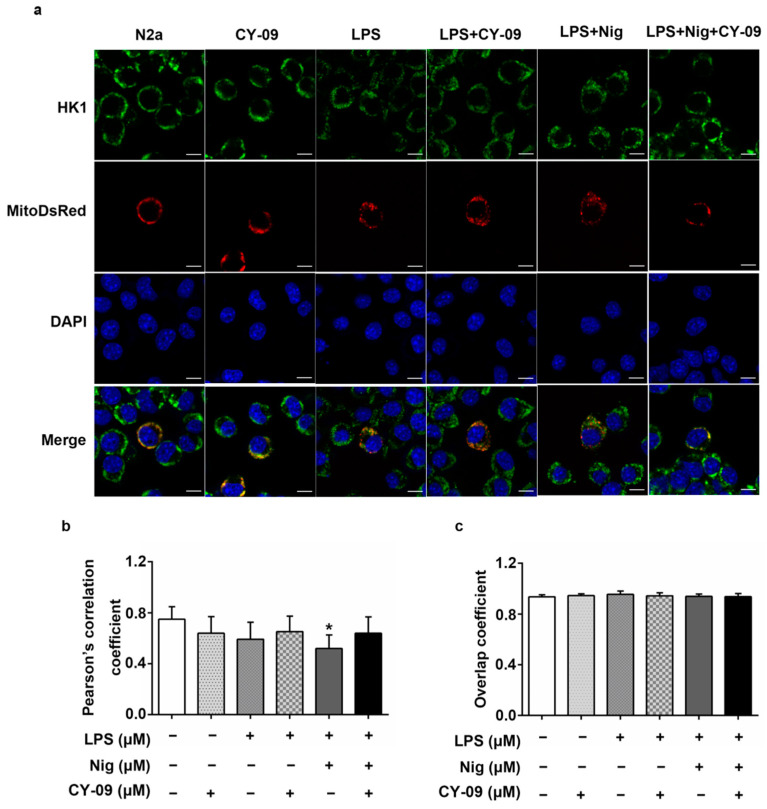
Distribution of HK1 in LPS-, Nig- and CY-09-treated N2a cells. (**a**) Immunofluorescence staining of HK1; Mito-DsRed-transfected N2a cells and treated N2a cells were imaged. DAPI is a nuclear stain. (**b**,**c**) Co-localization analysis of HK1 with mitochondria in cells. Pearson’s correlation and overlap coefficients were analyzed (*n* = 6, mean ± SD, Student’s *t*-test, * *p* < 0.05 vs. N2a cells. “−” means not added in the treatment. “+” means added in the treatment).

**Figure 10 ijms-22-01306-f010:**
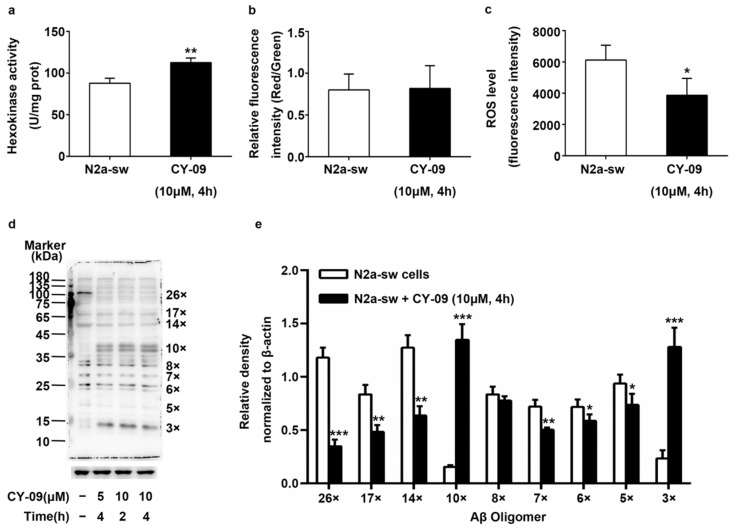
Detection of hexokinase activity, mitochondrial membrane potential, reactive oxygen species (ROS) levels and Aβ oligomers expression in CY-09-treated N2a-sw cells. (**a**) Hexokinase activity in N2a-sw cells and N2a-sw cells treated with 10 µM CY-09 for 4 h. (*n* = 3, mean ± SD, Student’s *t*-test, ** *p* < 0.01 vs. untreated N2a-sw cells). (**b**) Mitochondrial membrane potential in N2a-sw cells and N2a-sw cells treated with 10 µM CY-09 for 4 h. (*n* = 3, mean ± SD, Student’s *t*-test). (**c**) ROS levels in N2a-sw cells and N2a-sw cells treated with 10 µM CY-09 for 4 h. (*n* = 3, mean ± SD, Student’s *t*-test, * *p* < 0.05 vs. untreated N2a-sw cells). (**d**–**e**) Western blot analysis of Aβ oligomers in N2a-sw cells and CY-09 treated N2a-sw cells. (*n* = 3, mean ± SD, Student’s *t*-test; * *p* < 0.05, ** *p* < 0.01, *** *p* < 0.001 vs. untreated N2a-sw cells. “−” means no treatment).

## Data Availability

The data presented in this study are available upon request.

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
