# Peer review of "Effect of Increased IL-1β on Expression of HK in Alzheimer’s Disease"

_ijms, 2021, doi:10.3390/ijms22031306_

Round 1
Reviewer 1 Report
The manuscript by Han. S. et al., demonstrates a relationship between hexokinase (HK) and IL-1B in Alzheimer’s disease (AD) and suggests that inhibiting neuroinflammation may help improve neurodegenerative signs of AD. The authors show that decreased glucose metabolism is associated with decreased expression of HK in AD mice. They further show that the dissociation of HK1 from the mitochondria to cytosol happens in AD mice, which is associated with reduced glucose metabolism. Next, they show that the HK levels affect the levels of IL-1b and vice versa in a proportional manner, in neuroblastoma cell lines- n2a and n2a-sw, as a potential evidence of for targeting neuroinflammation for treatment of AD.
Overall, the study follows a good rationale for exploring the relationship between HK and neuroinflammation in AD by linking it to inflammasome activation. Nevertheless, study raises several important questions that needs to be addressed thoroughly.
Specific comments:
- The introduction is succinct but could benefit from more details about the rationale for selecting n2a vs n2a-sw cells and how this has not been shown previously or why it is important to use them for this study.
- HK1 is known to be a mitochondrial hexokinase isotype, the HK1 levels that are measured in 2.2 are cytosolic or mitochondrial fractions (in Fig. 2)?
- Statistical test is missing in Fig 2a. Authors should also clarify when they mention “students t-test”. Is it unpaired/paired; with Welch’s correction or not; any other correction methods have been applied or not!
- In 2.3 they show that mitochondrial detachment of HK1 by showing the significant difference b/n the cytosolic and mitochondrial fractions, which is the rationale for NLRP3 activation and downstream neuroinflammation. How does that correlate with data in 2.2?
- The most interesting part of the manuscript would be to know if this hexokinase activity is specific to Nlrp3 inflammasome or other inflammasomes are also involved (AIM2, NLRC4, Pyrin)?
- In Fig. 4a, what is the effect on Nlrp3 expression (protein and RNA)? Additionally, authors need to show Caspase-1 cleavage (pro- and cleaved form both) in addition to the IL-1b (both pro- and cleaved form). This is crucial to understand whether the effect is because of the direct activation of NLRP3 inflammasome or some other activation mechanism is involved.
- IL-1b secretion by ELISA must be shown to confirm/correlate the findings. All the blots should show pro and cleaved IL-1b (LPS+Nigericin is a well-known activator of Nlrp3 inflammasome).
- The HK1 measurements in the in vitro studies are mitochondrial or cytoplasmic fractions?
- Most in vitro treatments are only in N2a cells and not n2a-sw, why?
- In Fig. 5d cy-09 treatment shows an increase in IL-1b compared to control. Is this difference significant? Alternatively, would a caspase-1/Nlrp3 inhibitor be a better agent for IL-1b inhibition?
Reviewer 2 Report
In this study, Han and collaborators intended to establish an association between hexokinase dissociation from mitochondria and increased IL-1beta levels in Alzheimer’s disease (AD) pathology. Despite the scientific relevance of this study to the AD field, several points need to be addressed:
- Introduction and Discussion need to be significantly improved;
- Hexokinase activity should be determined in both in vivo and in vitro models of the disease;
- Figure 3 – very weak immunofluorescence signal of the mitochondrial network with Mito-DsRed;
- NLRP3 expression should be determine to support the working hypothesis of these authors;
- Figures 4f-g and 5f-g – modify the legend of the x-axis for an easier way to understand the different concentrations and exposure time;
- Immunofluorescence experiments colocalizing hexokinase and mitochondria should be performed for the experimental conditions using LND, LPS, Nig and CY-09;
- Are amyloid-beta levels affected in these experimental conditions?;
- In the last set of experiments, the authors should determine some mitochondrial-related parameters (mitochondrial potential, ROS,…) in N2a-sw cells in the presence of CY-09;
- A thorough review of the manuscript for English language is required.
Round 2
Reviewer 1 Report
The manuscript has been substantially improved. The only changes would be,
1. Authors mentioned that "LPS+Nigericin can activate other inflammasomes" is a wrong statement. Genetic studies using KO mice has shown that LPS+Nig /ATP specifically activates NLRP3 inflammasome whereas, Salmonella flagellin, dsDNA, and clostridium toxins are required for activation of NLRC4, AIM2 and Pyrin inflammasome respectively. If the authors have explored this point, the manuscript would definitely gain more attention.
Reviewer 2 Report
Overall, the authors have satisfactorily addressed most of the comments.
Author Response
Thank you very much for the reviews! Best Regards, the authors.